# Air Pollution and Respiratory System Responses in Healthy Adults Engaging in Outdoor Physical Exercise in Urban Environments: A Scoping Review

**DOI:** 10.3390/ijerph22091347

**Published:** 2025-08-28

**Authors:** Sergio Leonardo Cortés González, Katy Alexandra López Pereira

**Affiliations:** Physiotherapy Department, Faculty of Health, Pampalinda Campus, Universidad Santiago de Cali, Cali 760041, Colombia; kalopezp@unal.edu.co

**Keywords:** air pollution, respiratory tract, physical activity, respiratory diseases

## Abstract

Introduction: People who exercise outdoors in urban environments may inhale increased amounts of polluted air due to temporary respiratory changes induced by physical activity. The objective of this scoping review was to map the physiological, morphological, and/or functional responses of the respiratory system to air pollution in healthy adults who exercise outdoors in urban environments. Methods: This review was conducted following the guidelines of the Preferred Reporting Items Extension for Scoping Reviews (PRISMA-ScR). A comprehensive search of Medline (PubMed), Redalyc, Scielo, and Web of Science was conducted to identify clinical trials, quasi-experimental studies, and cross-sectional studies published in the last 10 years in English. Studies with healthy adult participants engaged in outdoor physical activity in urban environments were included. Texts with participants with preexisting respiratory diseases, elite athletes, animal models, and computer simulations were excluded. Results: The most frequently reported air pollutants were PM_2.5_, PM_10_, and ozone (O_3_); the most common forms of exercise were walking, running, and cycling. Exposure to air pollutants during physical activity was associated with reductions in forced vital capacity (FVC) and forced expiratory volume in 1 second (FEV_1_), as well as increases in the fraction of exhaled nitric oxide (FeNO) and proinflammatory biomarkers. Conclusion: The findings indicated that there are modifications in lung function in those who exercise outdoors. However, the association between these respiratory responses and air pollution was not statistically significant in most cases. Some authors suggested that the health benefits of physical activity could mitigate the harmful effects of air pollution.

## 1. Introduction

Air pollution (AP)-related mortality occurs primarily due to cardiorespiratory diseases. A study conducted across 652 cities found that exposure to air pollution was associated with a 0.36% increase in daily cardiovascular mortality and a 0.47% increase in respiratory mortality, especially among adults aged 35 to 70 years. These outcomes may be linked to the potential of AP exposure to alter the human respiratory microbiome, increasing susceptibility to viral infections [1]. According to the Global Burden of Disease study, air pollution rose from the 13th leading risk factor in 1990 to the 7th in 2019 in terms of disability-adjusted life years, accounting for approximately 6.67 million pollution-related deaths worldwide [2].

The high pollution levels observed in urban areas are mainly attributable to the increasing number of motor vehicles over the past three decades in major cities [3,4]. The combustion of fossil fuels, combined with environmentally inefficient industrial plants, generates various pollutants [3,4,5], including particulate matter (PM). PM refers to solid or liquid particles suspended in the air, with aerodynamic diameters ranging from 0.001 μm to 100 μm, originating from both natural and anthropogenic sources. These particles disrupt atmospheric radiation and cause harm to the human body when inhaled [5]. The most studied PM types in relation to human health are PM_10_ (particles ≤ 10 μm) and PM_2.5_ (particles ≤ 2.5 μm) [1].

Inhalation of PM and other pollutants such as carbon monoxide (CO), nitrogen dioxide (NO_2_), and ozone (O_3_) has been associated with respiratory conditions, including pneumonia, bronchitis, chronic obstructive pulmonary disease (COPD), and lung cancer [3,6]. Given the functional impairments these pollutants cause and the recurrent high-pollution episodes reported in both developed and developing countries, increasing attention has been paid in recent years to the impact of air pollution on health [7].

On another front, the World Health Organization (WHO) reported that as of December 2020, cardiovascular diseases and other non-communicable diseases (NCDs) were the leading causes of death and disability globally. These are largely attributable to high levels of physical inactivity and unhealthy lifestyles, in addition to genetic factors [8,9]. Sedentary behavior may be related to socio-economic, technological, and urban development, as environmental factors such as traffic congestion, urban violence, and lack of safe public spaces for physical activity (PA) tend to discourage outdoor exercise and limit access to its physical and mental health benefits [10,11].

In response, the WHO has called for increased global efforts in the prevention and management of NCDs, promoting PA as a key preventive strategy [8,9]. Together with the Pan American Health Organization, the WHO has issued a set of recommendations, including the adaptation of public spaces to support the greater use of active transport and the creation of safe and inclusive environments for PA accessible to all population groups, regardless of socioeconomic status [12].

However, when considering both the health risks posed by AP and the WHO’s recommendations for promoting PA, concerns arise regarding the potential respiratory effects of outdoor exercise in polluted environments. During PA, ventilation increases, potentially allowing a greater volume of pollutants to enter the respiratory system [2,7,11,13]. Studies involving children have shown that those living in high-traffic areas experience higher rates of coughing, wheezing, nasal discharge, and asthma, as well as impaired overall growth and reduced respiratory function, compared to children living in less polluted areas [14]. Nonetheless, evidence remains limited regarding the effects of such exposure in healthy adult populations.

A preliminary search in Google Scholar and PubMed identified studies examining the relationship between AP and respiratory diseases, such as asthma [15,16,17] and COPD [18], in relation to short- and long-term exposure during daily or occupational activities. Some studies were also found involving professional athletes; however, elite athletes demonstrate distinct cardiopulmonary adaptations—such as increased minute ventilation, enhanced alveolar diffusion capacity, and altered inflammatory responses—limiting their comparability to the general healthy adult population [19]. Similarly, occupational exposures are often chronic and highly specific, introducing additional sources of variability. To ensure consistency and relevance in the interpretation of findings, these groups were excluded from the present review.

Therefore, this review focuses specifically on the respiratory responses of healthy adults who regularly perform outdoor PA in urban environments under AP exposure.

Thus, this scoping review aims to map the physiological, morphological, and/or functional responses of the respiratory system to AP in healthy adults engaging in outdoor exercise in urban environments, as reported in scientific literature. As a secondary objective, this review also seeks to identify any reported responses in other organ systems resulting from the same exposure.

## 2. Methodology

A literature review was conducted on scientific articles published in English and Spanish between 2004 and 2024 in indexed journals. To ensure the most accurate information possible, studies from any country were included if they employed cross-sectional designs, clinical trials, quasi-experimental or experimental studies, case reports, or case-control studies. Gray literature and blog sources were excluded.

To enhance sample homogeneity and strengthen the validity of the conclusions, studies were included if participants were healthy adults over 18 years of age who regularly engaged in physical activity and/or outdoor exercise in urban environments—at least three times per week—across any exercise modality (e.g., walking, jogging, running, calisthenics, cycling, etc.). As previously established, due to the distinct physiological characteristics of elite athletes and the greater variability associated with occupational exposure, studies were excluded if participants were professional or elite athletes or if their physical activity was limited to occupational or routine daily tasks.

Additionally, studies involving participants diagnosed with respiratory diseases due to occupational exposure to chemicals, exposure to biomass or other biological contaminants, or genetic conditions were excluded.

Included studies had to report morphological or physiological findings related to the respiratory system, pulmonary function, and/or respiratory conditions diagnosed through clinical or diagnostic tests or reported by researchers using participant questionnaires. These findings had to be measured after outdoor PA following short-, medium-, or long-term exposure to AP. Studies involving animal models or computer simulations were excluded.

This review was conducted following the methodology outlined in the Preferred Reporting Items for Systematic Reviews and Meta-Analyses Extension for Scoping Reviews (PRISMA-ScR) [20]. The literature search was carried out using the following databases: Medline (PubMed) [21], Redalyc [22], Scielo [23], and Web of Science [24]. The search strategy used a combination of controlled vocabulary terms from Medical Subject Headings (MeSH), combined with free-text terms in both English and Spanish (see Appendix A). While the exclusion of studies in other languages may introduce language bias, English and Spanish were prioritized to ensure accessibility and methodological consistency.

After conducting the search, duplicates were removed using the Rayyan online platform. Once duplicates were eliminated, two reviewers independently and blindly screened the titles and abstracts using Rayyan, based on the predefined inclusion and exclusion criteria. To assess study quality, each reviewer independently and blindly applied the appropriate Joanna Briggs Institute (JBI) critical appraisal checklist for the study designs [25]. Studies with ≥60% of positively rated items were included (see Appendix A). While JBI does not specify an official threshold, this cutoff was adopted by consensus as a pragmatic response to the limited available evidence. This threshold is acknowledged as arbitrary and non-normative; therefore, future reviews are encouraged to use design-specific thresholds and more stringent criteria, particularly in contexts where evidence accuracy is critical.

Discrepancies between reviewers were resolved through consensus. The overall selection process is summarized in Figure 1, based on the PRISMA flowchart.

Data extraction was carried out using a Microsoft Excel spreadsheet, where information such as publication, authors, study design, population characteristics, pollutants analyzed, methods for evaluating the respiratory system, and main findings were recorded. After evaluating the initial design of the extraction matrix, additional fields were included for the study objectives and the methods used to measure air pollution to facilitate interpretation and analysis. If the method used to assess any respiratory system variable was unclear, the findings related to that variable were excluded. For the analysis of results, studies were initially grouped by study design. However, the synthesis of findings was later organized according to the types of respiratory responses reported.

## 3. Results

The aim of this scoping review was to map the available evidence regarding the physiological, morphological, and/or functional responses of the respiratory system to AP in healthy adults who engage in outdoor physical exercise in urban environments. After completing the identification and screening process, a total of eight (8) studies were included. All studies were conducted in outdoor urban environments, where physical activities such as walking, running, or cycling were performed, often during peak traffic hours in areas with high levels of AP. Except for one study, in which the participants were older adults over the age of 60 [26], the age of study participants ranged from 20 to 38 years [27,28,29,30,31,32,33], as shown in Table 1.

The most frequently reported pollutants were PM_2.5_ [26,27,28,29,31,32] and PM_10_ [26,27,28,29,30,31], although other pollutants such as black carbon (BC) or soot, NO_2_, and O_3_ [26,27,28,29,30,31,32,33] were also reported. These pollutants were measured using various specialized monitoring systems (see Appendix A). Primarily derived from vehicular traffic-related air pollution (TRAP), these airborne particles can easily enter the human respiratory tract due to their small size. Individuals who exercise near high-traffic areas are thus exposed to inhaling these pollutants [27], which may trigger mechanisms involving oxidative stress and inflammation as acute physiological responses. As noted by Sinharay et al. [26], “exposure of healthy individuals exercising to reconstituted diesel exhaust particles induces a broncho constrictive response with increased arterial stiffness and pulmonary inflammation”, suggesting that long-term exposure could be associated with increased respiratory morbidity and mortality.

The most reported outcomes were changes in lung function, assessed by spirometry, particularly in forced vital capacity (FVC) [26,27,28,30,31], forced expiratory volume in one second (FEV1) [26,27,28,30,31,32], and peak expiratory flow (PEF) [30,31,32]. Table 2 presents a synthesis of the studies grouped according to common exposure characteristics and outcome measures.

Studies by Matt et al. [27] and Kokot and Zejda [28] had similar objectives: to determine the acute effects (within 24 h post-exposure) of AP on lung function in healthy individuals performing PA, using statistical association models. Both studies reported that exposure to high pollution levels for 15 min led to a decrease in the FEV1/FVC ratio. This reduction was attributed to PM_2.5_, which showed similar average concentrations in both investigations.

According to Matt et al. [27], exposure to PM_2.5_ was associated with reductions in FEV1 (−1.31 mL·m^3^/μg, *p* = 0.02) and FVC (−1.71 mL·m^3^/μg, *p* = 0.02) in immediate post-exercise assessments. Similarly, Kokot and Zejda [28] reported that PM_2.5_ was an explanatory variable for the post-exercise decline in the FEV1/FVC ratio (OR: 1.03; 95% confidence interval: 1.00–1.06). However, while Matt et al. [27] acknowledged a reduction in pulmonary function, they also observed immediate respiratory benefits linked to PA. In contrast, Kokot and Zejda [28] suggested that acute changes in airflow following exercise under pollution exposure were influenced by ambient pollutant concentrations, particularly in healthy young adults.

Conversely, findings by Sinharay et al. [26] revealed that FEV1 increased following one hour of exercise, with improvements of 7.6% recorded between five and six hours post-exercise and a sustained increase of 3.6% after 26 h. These results suggest a protective or beneficial effect of aerobic PA on pulmonary function, which may outweigh the negative impact of AP exposure. Nonetheless, the authors noted that the observed improvements were attenuated in polluted environments. It is worth highlighting that this study was conducted among individuals over 60 years of age performing low- to moderate-intensity exercise (3–6 METs or 50–70% of maximum heart rate), a factor that may account for the divergence in pulmonary responses observed across the reviewed studies.

In addition to functional respiratory outcomes, inflammatory biomarkers were also reported as indicators of physiological response to AP. Pagani et al. [29] and Marmett et al. [33] observed elevated interleukin levels in participants who engaged in outdoor PA in areas with elevated concentrations of NO_2_ and O_3_, respectively. Marmett et al. [33] posited that PA could enhance ozone uptake, potentially blunting some of the physiological adaptations typically induced by exercise. This aligns with findings by Pagani et al. [29], who noted that elevated inflammatory markers persisted for up to 10 weeks following exposure. Additionally, FeNO levels—a marker of airway inflammation—were significantly decreased in runners’ post-exposure to high PM concentrations (*p* = 0.025), suggesting that endurance training may induce a distinct immunological response in the upper airways of physically active individuals compared to their sedentary counterparts, following PM_2.5_ exposure. However, the intensity of PA was not clearly specified.

## 4. Discussion

The primary objective of this scoping review was to map the existing evidence regarding the respiratory system responses in healthy adults who engage in outdoor PA in urban environments exposed to AP. To that end, we analyzed morphological, physiological, and functional outcomes reported in literature. While ethical concerns and challenges in controlling multiple variables limited the number of studies investigating the effects of AP exposure during exercise [2,34], available findings suggested potential respiratory impacts of engaging in outdoor PA during periods of high pollution. This was particularly evident in studies involving healthy university-aged adults (20–30 years) under pollution levels around 60 µg/m^3^ of PM_2.5_ and 100 µg/m^3^ of PM_10_.

Exercise induces physiological changes such as increased ventilation, airflow, a switch from nasal to oral breathing, and altered pulmonary diffusion, all of which facilitate the transport of a greater volume of pollutants to the deeper airways, increasing health risks [10,35,36], even during submaximal efforts (e.g., minute ventilation = 35 L/min) [10]. According to other investigations, cycling for 90 min in an area with PM_2.5_ concentrations of 100 μg/m^3^ may lead to more harm than benefit, with acute declines in lung function and a heightened long-term risk of developing chronic respiratory diseases due to exposure during PA [37,38].

Nevertheless, the evidence on pulmonary function decline remains inconclusive. Some authors emphasize the lack of significant changes in FEV1 and FVC in healthy adults and suggest that regular aerobic exercise—though frequency, duration, and intensity are often not specified—could mitigate, reverse, or buffer the adverse effects of AP on the respiratory, cardiovascular, and metabolic systems [2,11,39]. As activity levels increase, particle deposition may decrease due to higher exhalation rates, along with improved mucociliary clearance and reduced airway inflammation [14,39]. This review highlighted a contrast between the findings of Sinharay et al. [26], who reported post-exercise improvements in FEV_1_ and FVC, and those of Matt et al. [27], who observed declines in these parameters. Several factors may explain this discrepancy. Sinharay et al. studied older adults performing moderate-intensity exercise, whereas Matt et al. investigated young adults engaged in brief, vigorous activity in areas with heavy traffic. Older adults may experience transient improvements in lung function during low- to moderate-intensity exercise, possibly due to relatively greater airway elasticity or a more pronounced adaptive response to mild exertion. Additionally, pollutant concentrations were higher in the urban setting of Matt et al., likely increasing respiratory load and adversely affecting spirometry outcomes. Differences in the timing of post-exercise measurements (immediate vs. delayed) may further contribute to variability, as some adverse effects could be short-lived and reversible. These findings underscore the importance of accounting for exposure context, baseline fitness, and study design when assessing respiratory responses to exercise in polluted environments.

Regarding physiological responses, inhalation of O_3_ during exercise has been shown to trigger inflammatory cytokine production and oxidative stress in the respiratory system, leading to severe damage [2,33]. O_3_ can penetrate the bloodstream through alveolar diffusion across the alveolar–capillary barrier, promoting systemic inflammation and hypercoagulability. This enables the pollutant to cross the alveolar–capillary barrier, reach systemic circulation, and affect other organs, such as the central nervous system via neuroinflammation [40]. It also increases airway reactivity, potentially exacerbating responses to ambient allergens [41].

Furthermore, studies indicate that regular exercisers may show increased FeNO levels only when exposed to high concentrations of AP [11]. Low NO_2_ levels in active individuals may not directly affect pulmonary function but could impact other systems. Elevated FeNO levels could reflect peroxynitrite formation, leading to lipid peroxidation and reduced nitric oxide (NO) bioavailability—diminishing its vasodilatory effects and potentially causing vascular dysfunction [34].

Pagani et al. [29], in turn, reported a sustained increase in IL-6 levels and other inflammatory biomarkers after 10 weeks of outdoor exercise under high PM_2.5_ and PM_10_ exposure. IL-6 is a key proinflammatory cytokine involved in the respiratory system’s response to environmental stressors, including air pollutants such as PM_2.5_ and O_3_. Persistently elevated IL-6 has been linked to airway inflammation and increased airflow resistance, potentially manifesting as reduced FEV_1_. Conversely, a post-exercise decrease in IL-6—especially under moderate pollution—could indicate reduced bronchial inflammation and improved lung function. This is consistent with evidence that regular physical activity may modulate immune function and lower systemic inflammation. However, the relationship between inflammatory biomarkers and spirometric outcomes remains insufficiently studied. Future research should jointly assess both types of data and compare results across individuals with higher fitness levels and exercise-induced adaptations.

As for morphological changes, no human studies directly investigating structural changes were identified in this review. However, animal models and computational simulations provided useful insights. In laboratory mice, airway permeability was found to be altered during exercise in polluted environments, suppressing both pulmonary and systemic inflammatory responses [2,11]. Nonetheless, exposure to PM_2.5_ has also been associated with localized bleeding, pus exudation, inflammatory cell infiltration, and mucosal damage in the bronchial tissue [39].

Secondary cardiovascular responses must also be considered. Several studies suggest that exercising in polluted outdoor environments may reduce the cardioprotective benefits of physical activity through mechanisms such as pulmonary inflammation, increased blood pressure, and particulate deposition in the lungs [11,42,43]. PM_2.5_ particles can enter the bloodstream, leading to endothelial dysfunction, vasoconstriction, hypertension, and platelet aggregation. Indirectly, they may reach organs like the brain, disrupting autonomic nervous system function and triggering arrhythmias [14,44,45]. Although physical activity may counteract some negative effects of pollution, this should not be assumed universally. It is essential to distinguish between acute effects—such as transient inflammation or bronchoconstriction after immediate exposure—and chronic effects that may accumulate with regular exercise in high-pollution settings. Marmett et al. [33] found sustained elevations in inflammatory biomarkers following activity under O_3_ exposure, while Pagani et al. [29] reported similar outcomes with prolonged PM_2.5_ exposure, even among trained individuals. These findings point to a potential dose–response relationship involving pollutant levels, exercise intensity, and exposure duration.

Therefore, while physical activity is widely beneficial, practicing it in heavily polluted urban areas may pose risks if not accompanied by mitigation strategies—such as selecting cleaner routes, training during low-pollution hours, or using personal air quality monitors. Public health policies should incorporate these considerations, acknowledging environmental pollution as a key determinant of health.

Taken together, these findings suggest that AP-induced changes in the respiratory system may pose a risk factor for individuals engaging in outdoor exercise, potentially discouraging participation in recreational or sporting activities [35]. Therefore, encouraging outdoor PA in urban settings must be accompanied by preventive strategies, since the health benefits of PA—even in highly polluted areas—can outweigh the risks [46]. Although the harmful effects of air pollutants are acknowledged, the protective effects of regular PA are significantly greater [34].

This review encountered some limitations. Most studies included aerobic exercise interventions, but few specified the intensity of activity, preventing conclusive associations between PA characteristics and the respiratory responses observed. Additionally, inconsistencies in how AP levels were defined posed a challenge. Many studies used different thresholds to define “high pollution” and did not consistently compare their measurements with WHO standards, limiting the ability to assess statistical significance across findings.

For future research, we recommend that studies evaluating the health effects of outdoor exercise in polluted environments include precise descriptions of the exercise regimen—including intensity, duration, and modality—alongside pollution levels measured against WHO safety thresholds. This aligns with recommendations by Qin et al. [39], who proposed that moderate-intensity interval aerobic training may improve lung function and prevent tissue damage by mitigating oxidative stress and inflammation.

We recommend that future studies explore the long-term use of public spaces for physical activity, as current evidence on chronic respiratory responses to repeated exposure remains limited. Such research could enhance our understanding of cumulative health effects and inform the development of more effective public health interventions. Notably, substantial heterogeneity in study designs—including cross-sectional, crossover, and quasi-experimental methods—limits the comparability of findings and should be addressed in future research. Moreover, the lack of longitudinal studies and investigations into morphological changes in humans represents a significant gap in the existing evidence base.

A key limitation of this review is that most included studies were conducted in high-income countries such as the United States, Canada, the United Kingdom, Germany, and Australia. These studies typically involve homogeneous populations—healthy young adults with access to suitable urban environments for physical activity—which limits the generalizability of findings to other settings. In many low- and middle-income countries (LMICs), physical activity is often a necessity rather than recreational choices, such as walking or cycling for transport—frequently under high pollution exposure, with limited infrastructure and greater social vulnerability. More context-specific research in LMICs is urgently needed, using representative and culturally sensitive designs to better understand this interaction and inform inclusive, equitable public health policies.

A final limitation worth highlighting is that in studies where air quality measurement methods were not clearly specified, it was assumed that data were obtained from nearby local environmental monitoring stations. Although this assumption aligns with common practice in similar studies, we acknowledge that such inferences may introduce interpretative bias—particularly if the calibration, frequency, or reliability of the monitoring stations was not verified or reported. This methodological limitation could affect the accuracy of reported associations between exposure and physiological response, as pollutant concentrations may vary depending on distance, local meteorological conditions, or station type. Future studies are encouraged to clearly specify their environmental measurement methods to strengthen the internal validity of their findings.

In conclusion, individuals who frequently engage in outdoor PA in urban environments with high levels of AP—specifically PM_2.5_, PM_10_, O_3_, and NO_2_—may experience adverse respiratory responses, primarily affecting lung function parameters such as FEV1 and FVC. Additionally, physiological responses such as elevated levels of pro-inflammatory biomarkers have been observed. However, the current body of evidence is not conclusive, as many of these findings lack strong statistical significance.

Despite the presence of AP in urban outdoor settings, PA continues to offer clear health benefits for the prevention and management of NCDs. Literature suggests a potential protective effect of regular exercise, even in polluted environments, that may offset some of the negative respiratory outcomes associated with AP exposure.

## Figures and Tables

**Figure 1 ijerph-22-01347-f001:**
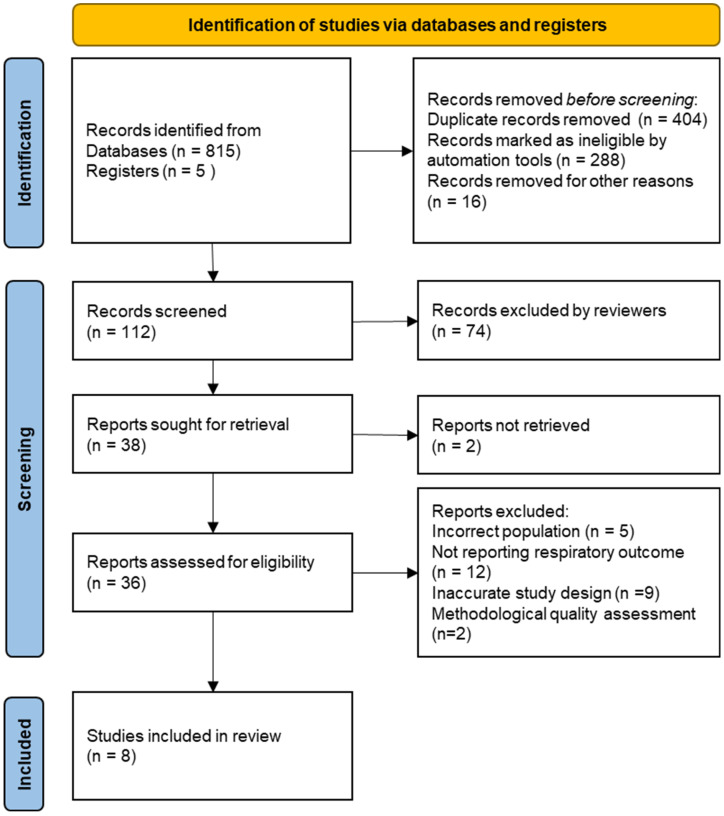
Flow diagram of study screening and selection process based on the Preferred Reporting Items for Systematic Reviews (PRISMA) guidelines. Source: authors’ own elaboration.

**Table 1 ijerph-22-01347-t001:** Characteristics of the included studies.

Author and Publication Year	Study Design	Population	Mean Age (Years)
Sinharay, R., et al. 2018 [26]	Randomized crossover study	135 participants	61.8
Matt, F., et al. 2016 [27]	Controlled crossover study	30 adults	36.0
Kocot, K., Zejda, J.E. 2020 [28]	Controlled clinical trial	30 young adults	22.5
Pagani, L.G., et al. 2020 [29]	Quasi-experimental study	28 adults	37.4
Strak, M., et al. 2010 [30]	Quasi-experimental study	12 adults	30.0
Kocot, K., et al. 2021 [31]	Quasi-experimental study	71 young adults	20.7
Kesavachandran, C.N., et al. 2015 [32]	Cross-sectional analytical study	378 adults	30.6 ± 10.6
Marmett, B., 2023 [33]	Cross-sectional study	100 adult males	25.5 ± 6.2

Source: authors’ own elaboration.

**Table 2 ijerph-22-01347-t002:** Summary of results.

Respiratory System Outcome	Authors	Exposure Time	Reported Air Pollutants	Main Results	Statistical Significance
Lung Function (FEV1, FVC, PEF)	Matt, F., et al. [27]	15 min	NO_x_, NO, BC, UP, PM_2.5,_ PM_10,_ Coarse PM	Decrease in FEV1 and FVC post-exercise. Increase in HRmax reduced the immediate negative effects on PEF and the delayed effects on FVC.	*p* = 0.02
	Kocot, K., Zejda, J.E. [28]	15 min	NO_x_, PM_2.5,_ PM_10,_ SO_2_	Decrease in FEV1/FVC post-exercise. SO_2_ was a predictor of airflow reduction.	OR: 1.03 (IC95%: 1.00–1.06)
	Strak, M., et al. [30]	45–60 min	PM_10,_ BC, PNC	Decrease in FVC, PEF, and FEV1 six hours post-exercise.	*p* = 0.04
	Kocot, K., et al. [31]	45–60 min	PM_10,_ PM_2.5_, SO_2_, NO_2_	FVC decreased post-exercise, and FEV1_ref_ decreased during exposure.	*p* = 0.02
	Kesavachandran, C.N., et al. [32]	45–60 min	PM_10_, PM_2.5_	Decrease in FEV1 and PEF in those who do not exercise.	*p* = 0.001
	Sinharay, R., et al. [26]	1 h	BC, NO_2_, PM_10_, PM _2.5_, UP	FEV1 increased post-exercise. Increase in FVC at 3 h and 5 h post-exercise.	*p* < 0.05
Inflammatory Biomarkers	Pagani, L.G., et al. [29]	Not specified	PM_2.5_, PM_10_	IL-6 levels increased. After 10 weeks, IL-10 increased in sedentary individuals, and IL-17A in runners.	*p* = 0.025
	Marmett, B., et al. [33]	Not specified	O_3,_ NO_2_	High levels of IL-1β, IL-6, IL-10 in individuals with low physical fitness.	*p* = 0.01
FeNO	Pagani, L.G., et al. [29]	Not specified	PM_2.5,_ PM_10_	Decrease in FeNO after 10 weeks of exposure.	*p* = 0.025

Abbreviations: BC = black carbon, NO_2_ = nitrogen dioxide, UP = ultrafine particles, PM = particulate matter, FeNO = fractional exhaled nitric oxide, FEV1 = forced expiratory volume in 1 second, FVC = forced vital capacity, PEF = peak expiratory flow, NO = nitric oxide, NO_x_ = nitrogen oxides, SO_2_ = sulfur dioxide, O_3_ = ozone, PNC = particle number concentration, HRmax = maximum heart rate, IL = interleukins. Source: authors’ own elaboration.

## Data Availability

No new data were created or analyzed in this study. Appendix A is available from the corresponding author upon reasonable request.

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
