# Peer review of "Air Pollution and Respiratory System Responses in Healthy Adults Engaging in Outdoor Physical Exercise in Urban Environments: A Scoping Review"

_ijerph, 2025, doi:10.3390/ijerph22091347_

Round 1

Reviewer 1 Report

Comments and Suggestions for Authors

This paper covers an important intersection between urban air pollution and public health recommendations for physical activity. The paper is well written and interesting to read, however I see the following issues that should be resolved before publishing this paper.

  1. Authors mention "not statistically significant” Please clarity on statistical significance ans discuss statistical power, sample size limitations, and variability across exercise protocols to help readers interpret this outcome.
  2. Most studies come from high-income countries and feature relatively homogeneous populations (young adults). please add information of the geographic limitations and emphasize the need for studies in low- and middle-income regions, where outdoor pollution and physical activity intersect differently.

Author Response

Cover Letter – Response to Reviewers

Dear Editors,

We are grateful for the opportunity to resubmit our manuscript entitled “Air Pollution and Respiratory System Responses in Healthy Adults Engaging in Outdoor Physical Exercise in Urban Environments: A Scoping Review.” We thank the reviewers and editorial team for their thoughtful and constructive feedback, which has substantially improved the clarity and rigor of our work.

Please find below our detailed, point-by-point responses to the reviewers’ comments. Modifications in the revised manuscript are highlighted in green (Reviewer 1) and yellow (Reviewer 2), as requested. Line numbers refer to the updated version of the manuscript.

Reviewer 1 – Responses (highlighted in green)

  • Clarify the predominance of studies from high-income countries and limitations in generalizability.
    Response: Addressed in the Limitations section (Lines 348–357), emphasizing the need for more context-sensitive research in LMICs.
  • "We also addressed redundancy by removing the concurrent use of the phrase 'statistically significant' alongside the reported p-value."

We thank you again for the insightful comments and your continued support. We hope the revised manuscript now meets the standards required for publication and are happy to provide any further clarification or information if needed.

Sincerely,

Authors:

Sergio Leonardo Cortes González

Katy Alexandra López Pereira

Reviewer 2 Report

Comments and Suggestions for Authors

  • The primary and secondary objectives are clearly stated, but the rationale for excluding elite athletes and occupational exposures could be expanded. While the distinction is noted, a brief justification (e.g., physiological differences or exposure variability) would bolster the manuscript’s focus on "healthy adults."
  • The secondary objective (other organ systems) is mentioned but not systematically addressed in results/discussion. Either integrate these findings or clarify their exclusion.
  • The databases selected (PubMed, Redalyc, Scielo, Web of Science) are appropriate, but the exclusion of non-English/Spanish studies should be justified given the global nature of the topic. Potential language bias should be acknowledged.
  • The threshold of "≥60% positive responses" in the JBI checklist is arbitrary. Provide a rationale for this cutoff or reference established guidelines.
  • The assumption that unclear methods default to local monitoring stations may introduce bias. Studies with poorly described methods should be flagged as a limitation.
  • Table 2 is dense and could benefit from reorganization (e.g., grouping by pollutant type or response category). Include p-values or effect sizes where available to highlight statistical significance.
  • The contrast between Sinharay et al. (FEV1 improvement) and Matt et al. (FEV1 decline) is noted but not critically analyzed. Discuss potential moderators (e.g., exercise intensity, age, pollution levels) to reconcile these differences.
  • The discussion of oxidative stress and inflammation is thorough, but link these mechanisms more explicitly to the observed spirometric changes (e.g., how might IL-6 reductions correlate with FEV1?).
  • The conclusion that PA benefits may offset AP harms is plausible but overly generalized. Differentiate between acute vs. chronic exercise effects and consider dose-response relationships (e.g., Marmett et al.’s O3 findings vs. Pagani et al.’s PM2.5 results).
  • The manuscript acknowledges variability in pollution thresholds and exercise intensity but does not address heterogeneity in study designs (e.g., crossover vs. longitudinal). Discuss how this impacts comparability.
  • Highlight the predominance of short-term exposure studies and the lack of morphological data in humans as key evidence gaps.
  • Define abbreviations at first use (e.g., TRAP in Line 156).
  • Clarify "low-to-moderate intensity exercise" (Page 6) with examples (e.g., METs, heart rate ranges).
  • Figure 1: Ensure the PRISMA flowchart matches the text (e.g., "n=8" studies included vs. "n=9" in Table 1).
  • Cite all annex tables in the main text (e.g., Annex 1 is referenced as "Appendix I" in Methods).
  • Avoid redundant phrasing (e.g., "statistically significant" and "p=0.02" in Lines 175–177).
  • Page 8, Line 236: "O3 can penetrate the bloodstream" → Specify inhalation-to-bloodstream pathways (e.g., alveolar diffusion).

Comments on the Quality of English Language

 The English could be improved to more clearly express the research.

Author Response

Cover Letter – Response to Reviewers

Dear Editors,

We are grateful for the opportunity to resubmit our manuscript entitled “Air Pollution and Respiratory System Responses in Healthy Adults Engaging in Outdoor Physical Exercise in Urban Environments: A Scoping Review.” We thank the reviewers and editorial team for their thoughtful and constructive feedback, which has substantially improved the clarity and rigor of our work.

Please find below our detailed, point-by-point responses to the reviewers’ comments. Modifications in the revised manuscript are highlighted in green (Reviewer 1) and yellow (Reviewer 2), as requested. Line numbers refer to the updated version of the manuscript.

Reviewer 2 – Responses (highlighted in yellow)

  1. Justify the exclusion of elite athletes and occupational exposures.
    Response: This has been clarified in the Introduction (Lines 82–88) and Methodology (Lines 102–109).
  2. Acknowledge language bias due to exclusion of non-English/Spanish studies.
    Response: Added to Methodology (Lines 124–126).
  3. Justify the ≥60% threshold for JBI checklist.
    Response: Explained as a consensus-based decision, with recommendations for more rigorous thresholds in future research (Lines 132–136).
  4. Consider poorly described pollution measures as a limitation.
    Response: Discussed in detail in the Discussion section (Lines 359–368).
  5. Reorganize Table 2 and include p-values/effect sizes where available.
    Response: Table 2 has been revised for clarity; additional data included where applicable (Line 217).
  6. Explain differing FEV₁ findings (Sinharay vs. Matt).
    Response: A paragraph comparing study contexts, exercise intensity, and pollution levels has been added (Lines 247–261).
  7. Discuss correlation between IL-6 reductions and FEV₁.
    Response: Relationship explored and discussed with reference to exercise-induced modulation (Lines 280–289).
  8. Differentiate acute vs. chronic effects and highlight dose–response aspects.
    Response: Discussion expanded to emphasize these distinctions and their implications (Lines 304–317).
  9. Acknowledge heterogeneity in study designs.
    Response: Noted in the Discussion as a limitation affecting comparability (Lines 343–345).
  10. Emphasize lack of longitudinal data and morphological outcomes.
    Response: Identified as key evidence gaps (Lines 345–347).
  11. Define abbreviations (e.g., TRAP).
    Response: Defined upon first use (Line 167).
  12. Clarify “low-to-moderate intensity exercise.”
    Response: Now defined using METs and heart rate ranges (Line 202).
  13. Ensure consistency in PRISMA diagram and text.
    Response: Confirmed that 8 studies were included; clarified in the text (Line 156).
  14. Cite all appendices in main text.
    Response: We respectfully request that this appendix not be integrated into the main text, as it was meant only to illustrate the process, not to contribute substantively. A note has been added to clarify this.
  15. Avoid redundancy in statistical expressions.
    Response: Edited to eliminate redundancy between text and p-values (Lines 186–190).
  16. Especifique cómo el O₃ entra al torrente sanguíneo.
    Respuesta: Ahora se describe como «difusión alveolar a través de la barrera alveolocapilar» (líneas 265-266).

Agradecemos nuevamente sus valiosos comentarios y su continuo apoyo. Esperamos que el manuscrito revisado cumpla con los estándares requeridos para su publicación y estaremos encantados de proporcionar cualquier aclaración o información adicional si es necesario.

Atentamente,

Autores:

Sergio Leonardo Cortés González

Katy Alexandra López Pereira

Round 2

Reviewer 2 Report

Comments and Suggestions for Authors

the article is suitable for publication as the authors have made the necessary edits